# Effects of nurse-led transitional care interventions for patients with heart failure on healthcare utilization: A meta-analysis of randomized controlled trials

Minlu Li[1,2☯], Yuan Li[1☯], Qingtong Meng[1¤], Yinyin Li[1], Xiaomeng Tian[3], Ruixia Liu[1], Jinbo Fang[1] *

1 West China School of Nursing, Sichuan University, Chengdu, Sichuan, China, 2 Department of Neurology, West China Hospital, Sichuan University, Chengdu, Sichuan, China, 3 Department of Cardiology, West China Hospital, Sichuan University, Chengdu, Sichuan, China

☯ These authors contributed equally to this work.
¤ Current address: Department of Cardiology, Shenzhen People's Hospital, Shenzhen, Guangdong, China
* fangjinbo1107@163.com

**Data Availability Statement:** All relevant data are within the paper and its Supporting Information files.

## Abstract

### Background

Heart failure (HF) imposes a substantial burden on patients and healthcare systems. Hospital-to-home transitional care, involving time-limited interventions delivered predominantly by nurses, was introduced to lighten this burden. This study aimed to examine the effectiveness and dose-response of nurse-led transitional care interventions (TCIs) on healthcare utilization among patients with HF.

### Methods

Health-related databases were systematically searched for articles published from January 2000 to June 2020. We included randomized controlled trials (RCTs) that compared nurse-led TCIs with usual care for adults hospitalized with HF and reported the following healthcare utilization outcomes: all-cause readmissions, HF-specific readmissions, emergency department visits, or length of hospital stay. Random-effects meta-analysis, meta-regression analysis, and dose-response analysis were performed to estimate the treatment effects and explain the heterogeneity.

### Results

Twenty-five RCTs including 8422 patients with HF were included. Nurse-led TCIs for patients with HF resulted in a mean 9% (RR = 0.91; 95% CI = 0.82 to 0.99; p = 0.04; $I^2$ = 46%) and 29% (RR = 0.71; 95% CI = 0.60 to 0.84; p < 0.0001; $I^2$ = 0%) reduction in all-cause and HF-specific readmission risks respectively compared to usual care. The interventions were also effective in shortening the length of hospital stay (MD = -2.37; 95% CI = -3.16 to -1.58; p < 0.0001; $I^2$ = 14%). However, no significant reduction was found for emergency department visits (RR = 0.96; 95% CI = 0.84 to 1.10; p = 0.58; $I^2$ = 0%). The effect of

**Funding:** This work was supported by the Key Project of Science and Technology of Sichuan Province, China (Grant No. 2020YFS0150) and the West China Nursing Discipline Development Special Fund, Sichuan University (Grant No. HXHL19024). The funders had no role in study design, data collection and analysis, decision to publish, or preparation of the manuscript.

**Competing interests:** The authors have declared that no competing interests exist.

meta-regression coefficients on all-cause and HF-specific readmissions was not statistically significant for any prespecified trial-level characteristic. Dose-response analysis revealed that the HF-specific readmission risk decreased in a dose-dependent manner with the complexity and intensity of nurse-led TCIs.

## Conclusions

Nurse-led TCIs were effective in decreasing all-cause and HF-specific readmission risks, as well as in reducing the length of hospital stay; however, the interventions were not effective in reducing the frequency of emergency department visits.

## Introduction

Heart failure (HF) is a common clinical syndrome that imposes a substantial economic burden on global healthcare systems [1], of which approximately 80% is attributable to the high hospitalization and readmission rates [2]. It is well established that this patient population is extremely vulnerable in the immediate post-discharge period, with a 30-day readmission rate of 25% [3,4]. Approximately 40% of these early readmissions have been found to be preventable and relevant to suboptimal transitional care due to the short of care coordination and continuity for patients in transition between healthcare settings or providers [5,6]. Transitional care interventions (TCIs), defined as a broad range of time-limited actions offered to ensure health care continuity, have been implemented to interrupt the pattern of frequent use of healthcare services [1,5,6]. Although no consensus exists on the type of TCIs provided, classification of the various interventions employed for the transition of care, or length of the transitional care period, these interventions are generally aimed at improving patient outcomes through preplanned, preventive, and supportive care or close patient monitoring [7–9].

Nurses are well versed in disease management and self-management support, as well as in promoting self-care for people with chronic illness [10]. Classical models of care transitions have been established such as the Transitional Care Model [11], the Care Transitions Intervention Model [12], and the Guided Care Model [13]. All of those rely on nurses—especially those prepared for advanced practice—to assume pivotal roles in transitional care service delivery [14]. Numerous nurse-led TCIs have been thereafter implemented with the expectation of promoting healthy behavior and reducing healthcare utilization.

Previous systematic reviews and meta-analyses have indicated TCIs as an effective intervention strategy for improving health outcomes in patients with HF [8,15–18]. However, transitional care is often confused with the broader concept of chronic disease management in the literature, because studies have concentrated on the intervention effects without considering the time-limited nature of TCIs and the vulnerable period of hospital-to-home transition [8,15,16]. In addition, at the time of writing, no systematic review of randomized controlled trials (RCTs) exploring the healthcare utilization impact of nurse-led TCIs among patients with HF was available. Hence, we undertook the current systematic review and meta-analysis of RCTs comparing nurse-led TCIs with usual care to examine the intervention effects for patients with HF on healthcare utilization outcomes. In particular, we synthesized the most up-to-date evidence with the objective of assessing the effects on all-cause and HF-specific readmissions, emergency department (ED) visits, and length of hospital stay (LOS), and identifying potential trial-level characteristics that affect the overall effectiveness.

Furthermore, previous research has suggested that complex (i.e., combination of multiple intervention components) and high-intensity (i.e., frequent contacts of significant duration) interventions tended to attain better outcomes than those with medium- or low-intensity [15,17]. We thus hypothesized a dose-response relationship between the intensity and complexity of interventions and program efficacy, and utilized the dose-response meta-regression method to examine the hypothesis.

## Methods

The protocol of this systematic review was prospectively registered with the PROSPERO database (CRD42020202602). The study was reported in compliance with the PRISMA guidelines [19].

### Search strategy

A structured literature search was conducted using the MEDLINE, Embase, Cochrane Library, and CINAHL databases with no restrictions on language or publication status to identify all relevant articles published from 1 January 2000 to 31 June 2020. The bibliographies of all relevant articles were manually searched for additional eligible publications. A comprehensive search strategy using a combination of MeSH terms and free text words such as "heart failure," "hospital discharge," "patient follow-up," "coordinated care," "nurse-led," and "various types of TCIs" was developed with the assistance of an experienced medical librarian. The complete search strategies for all databases can be found in the S1 File.

### Inclusion and exclusion criteria

RCTs were included if they met the following criteria: a) recruited patients aged 18 years or over with a primary diagnosis of HF who were discharged from hospital to home; b) compared nurse-led TCIs that were initiated during or within 1 week of the index HF hospitalization with usual care; and c) reported at least one of the following healthcare utilization outcomes within a 6-month period from discharge: all-cause readmissions, HF-specific readmissions, ED visits, and hospital LOS. The 6-month timeline for outcome assessment was set out of consideration for the time-limited nature of TCIs and that outcomes far away from the index hospitalization probably reflect the natural history of HF or an unrelated illness [15].

We excluded studies that recruited patients with general cardiac disorders rather than HF specifically and those that focused on medical practices with nurses only assisting in parts of the interventions (i.e., nurses did not play the central role or had autonomous decision-making and authority in the team).

### Study selection and risk of bias assessment

First, the lead investigator excluded all duplicate articles. Thereafter, two independent investigators screened the titles and abstracts of the remaining articles for eligibility. Finally, full-text articles that potentially met our inclusion criteria were retrieved and scrutinized. The internal validity (risk of bias) of the included studies was evaluate independently by the same two investigators using the latest version of the Cochrane risk of bias tool for randomized trials (RoB 2) [20]. This tool consists of five bias domains addressing bias due to the randomization process, deviations from intended interventions, missing outcome data, measurement of the outcome, and selection of reported results. Trials were categorized as low risk of bias, some concerns, or high risk of bias within each domain and assigned an overall risk of bias judgment. Any

disagreements regarding the study selection or risk of bias assessment were resolved by discussion or adjudication by the lead investigator.

## Data extraction

Two investigators extracted data independently from each included RCT using a standardized data extraction form, which was similar to that developed in our previous review [7]. Data on the following items were extracted: study identifier, design, and setting; participant characteristics; description of interventions and comparators; assessment tool(s); quantitative outcomes; and items pertaining to the methodological soundness of the studies. Differences in data extraction were resolved by discussion and consensus.

## Data synthesis

All statistical analyses were performed using RevMan 5.4 (The Cochrane Collaboration, Oxford, UK) and STATA 15.1 (Stata Corp., College Station, TX, USA). The pooled risk ratio (RR) and mean difference (MD) with 95% confidence interval (CI) were reported as the effect size metrics. Statistically significant RRs were translated into numbers-needed-to-treat (NNT) to gauge their clinical relevance. Had cluster trials been included, we would adjust for clustering where the necessary data was provided, or else sensitivity analyses would ensue. The random-effects model was selected a priori because of the anticipated complexity and multicomponent nature of nurse-led TCIs. Meta-analyses were performed on an intention-to-treat basis where appropriate. Heterogeneity between studies was explored by visual inspection of forest plots and evaluation of the $I^2$ statistics. $I^2$ values of 75%, 50%, 25%, and 0% were considered high, moderate, low, and no heterogeneity, respectively [21]. To assess the robustness of the overall effect estimates, we carried out sensitivity analysis by excluding one study at a time during repeated analyses. Furthermore, publication bias was evaluated by visually inspecting funnel plots along with Harbord's modified tests when ten or more trials were pooled. If publication bias was suspected, the trim-and-fill procedure was conducted to estimate the influence of potentially missing studies on summary effect estimates [22].

In addition, we conducted meta-regression analysis to explore whether any trial-level covariate could explain the between-study heterogeneity, considering certain trial characteristics (such as study region, year of publication, and study quality), participant characteristics (such as age, six, left ventricular ejection fraction, and New York Heart Association functional classification), and intervention characteristics (such as communication method, intervention environment, involvement of caregiver, place of intervention initiated, and intervention length). We also employed dose-response analysis with the one-stage robust error meta-regression (REMR) approach [23] to explore the relationship of complexity and intensity of the interventions ("program dose") with treatment effects. The complexity hereby referred to the number of intervention components prescribed, and intensity referred to the frequency and duration of care contacts. The validated Heart Failure Disease Management Scoring Instrument (HF-DMSI) developed by Riegel et al. [24] was adapted to quantify the overall "program dose" of nurse-led TCIs provided in different studies. The summary score of the instrument ranges from 5 to 35 (S2 File), where higher scores indicate a more intensive and complex intervention program, that is, more healthcare contacts and intervention components were involved in the program [24].

## Quality of evidence

We applied the Grading of Recommendations Assessment, Development, and Evaluation (GRADE) guidelines [25] to assess the strength of evidence for each outcome. GRADEpro

GDT software was used to generate a "Summary of findings" table [26]. We determined whether to downgrade the quality level of evidence in compliance with the GRADE's criteria based on risk of bias, inconsistency, indirectness, imprecision, and publication bias. As such, the certainty of the evidence for each outcome was deemed to be high, moderate, low, or very low.

# Results

## Search results and characteristics of included studies

The results of the initial literature search and study selection are outlined in the PRISMA flow diagram (Fig 1). Of the 3157 articles identified, full texts of 136 articles were retrieved and evaluated for potential inclusion, of which 25 trials meeting the criteria were ultimately included

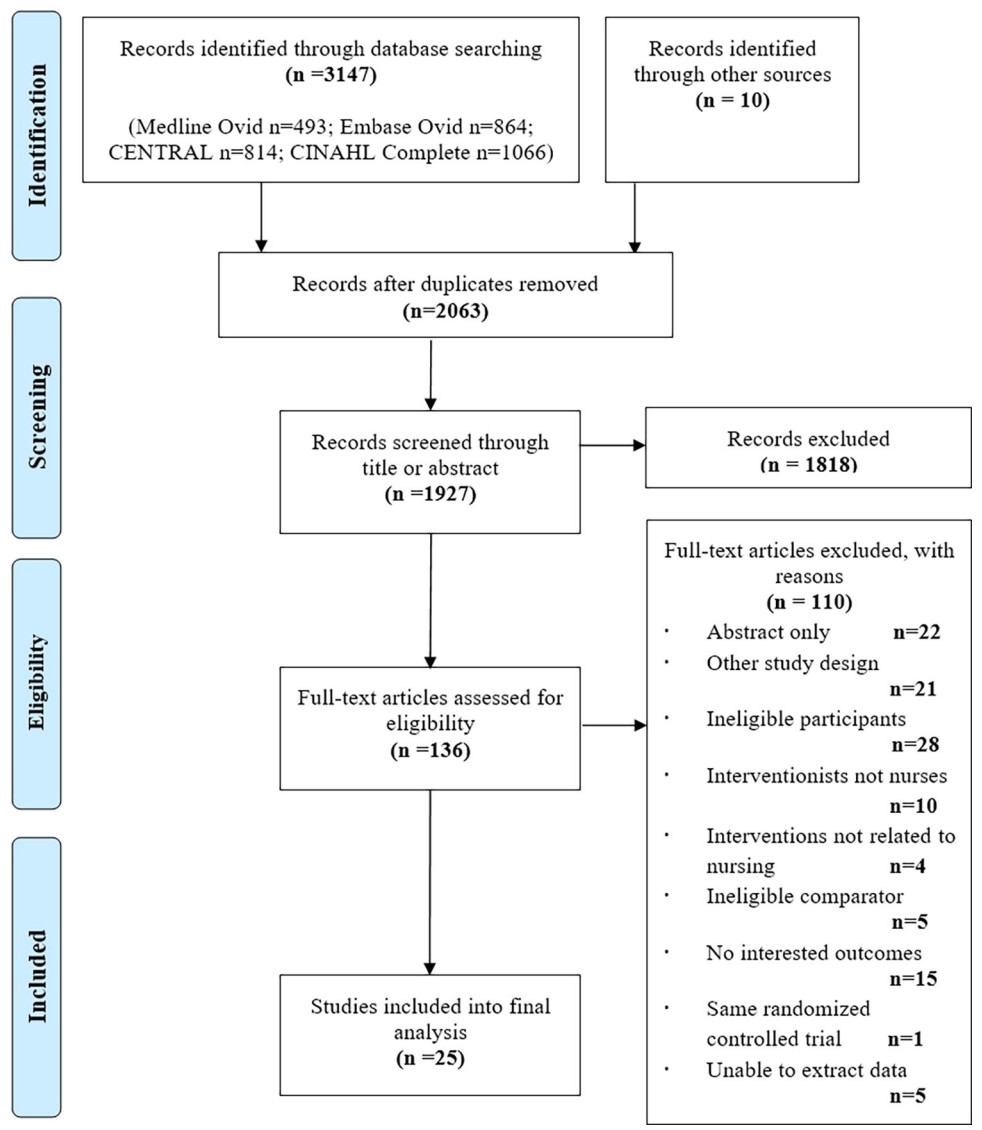

**Fig 1. Flowchart illustrating the search strategy (limit time 2000–2020).** *From*: Moher D, Liberati A, Tetzlaff J, Altman DG, The PRISMA Group (2009). Preferred Reporting Items for Systematic Reviews and Meta-Analyses: The PRISMA Statement. PLOS Med 6(7): e1000097. doi:10.1371/journal.pmed1000097.

and analyzed. Particularly, we included 22 individually randomized RCTs and 3 cluster RCTs, involving a total of 8422 participants from 12 countries (Table 1). Except for 3 studies conducted in low- and middle-income countries, all studies were conducted in high-income countries, with almost half of them were in the US [11,27–37]. The mean age of the study population ranged from 61.9 [29] to 77.5 years [38], and 39.6%~72.6% [39,40] of the participants were men. Nineteen trials measured the mean or median left ventricular ejection fraction, of which 10 reported a mean value of less than 40%; and 14 trials indicated HF severity based on the New York Heart Association classification.

The characteristics of the included studies are summarized in Table 1 (for further details, see S3 File). Multiple intervention categories were used in the included RCTs to develop transitional care strategies, of which structured telephone support [27,28,33,34,41–43] and case management [11,30–32,35,44,45] were the most widely used, and all intervention contents emphasized nurse-led close monitoring, education, counseling, and extended follow-up to ensure hospital-to-home healthcare continuity. Although there were some overlaps across studies in terms of interventions programs, active interventions varied considering the intervention categories, recipient points, intervention contents, delivery personnel, communication methods, duration, complexity and intervention environments (S3 File). The length of interventions ranged from 0.5 [39,46] to 6 months [11,28,29,31–34,40,41,44,47–49]. The overall "program dose" varied from 13 [46] to 31 [11] across different studies. In addition, most studies only briefly described usual care, which generally consisted of basic discharge education and routine follow-up with the primary care physician or cardiologist, as required.

## Risk of bias

The risk of bias of included trials was assessed using the RoB 2. Regarding the overall risk of bias, all RCTs, except five, had high risk of bias or had some concerns. We constructed "traffic-light" plots (Figs 2–5) using the online robvis tool to visualize the risk of bias assessments and judgment distribution within each domain for the included RCTs [20]. The main weakness of the included RCTs was related to random progress and allocation concealment. It was impossible to blind participants and interventionists owing to the nature of the interventions.

## Effects of nurse-led TCIs in patients with HF

**All-cause readmissions.**   Nineteen studies reported the outcome of all-cause readmissions (Fig 2). The combined data showed a significant reduction in readmission risk among patients who received nurse-led TCIs compared with that among those who only received usual care (RR = 0.91; 95% CI = 0.82 to 0.99; p = 0.04). Heterogeneity across studies was moderate ($I^2$ = 46%). The NNT was 27, meaning 27 patients had to receive interventions to prevent one patient from hospital readmission. Sensitivity analysis findings were similar to the main findings upon exclusion of each of the included studies from the analysis (S4 File); hence, the findings of this meta-analysis were robust.

**HF-specific readmissions.**   Data from 10 RCTs were pooled to assess the effect of nurse-led TCIs on HF-specific readmissions (Fig 3). The meta-analysis revealed that nurse-led TCIs resulted in a 29% reduction in the HF-specific readmission risks (RR = 0.71; 95% CI = 0.60 to 0.84; p < 0.0001), with no evidence of between-study heterogeneity ($I^2$ = 0%). The NNT to prevent one patient from hospital readmission was estimated to be 15. The results were robust with no substantial changes in sensitivity analyses (S4 File).

**Emergency department visits.**   Five RCTs reported ED visit data (Fig 4). The pooled analysis failed to detect statistically significant differences between nurse-led TCIs and usual care (RR = 0.96; 95% CI = 0.84 to 1.10; p = 0.58), with no heterogeneity across the studies ($I^2$ = 0%).

Table 1. Summary characteristics of participants and interventions.

| Author/Year/ Country | Trial characteristics | | Characteristics of participants | | | | Characteristics of interventions | | | | | | Adapted HF-DMSI[¶] |
|---|---|---|---|---|---|---|---|---|---|---|---|---|---|
| | Region[*] | Risk of bias | Age (years) | Male (%) | LVEF[†] (%) | NYHA[‡] III-IV (%) | Intervention category | Communication[§] | Environment | Caregiver | Place initiate | Duration of intervention (months) | |
| Aldamiz-Echevarria 2007 Spain | HIC | Some concern | 75.8 | 39.6 | ≥40 | NR | Home visiting program | F-to-F | Home-based | Yes | Community | 0.5 | 20 |
| Angermann 2012 Germany | HIC | Low | 67.7 | 71 | <40 | 40 | Structured telephone support | P-to-P | Telephone or internet-based | Yes | Hospital | 6 | 26 |
| Barth 2001 USA | HIC | High | 75.2 | 47.1 | NR | NR | Structured telephone support | P-to-P | Telephone or internet-based | No | Community | 2 | 16 |
| De Souza 2014 Brazil | LMIC | High | 62 | 62.7 | <40 | 55.9 | Home visiting program | Combined | Combination of settings | Yes | Community | 6 | 26 |
| Domingues 2011 Brazil | LMIC | High | 63 | 57.7 | <40 | NR | Structured telephone support | P-to-P | Telephone or internet-based | Yes | Community | 3 | 19 |
| Ducharme 2005 Canada | HIC | Some concern | 69 | 72 | <40 | 90.4 | Multidisciplinary care model | Combined | Combination of settings | Yes | Community | 6 | 28 |
| Dunagan 2005 USA | HIC | Some concern | 70.5 | 43.7 | <40 | 80.1 | Structured telephone support | P-to-P | Telephone or internet-based | No | Community | 6 | 17 |
| Kasper 2002 USA | HIC | Some concern | 61.9 | 60.5 | <40 | 58.5 | Clinic-based intervention | Combined | Combination of settings | No | Community | 6 | 26 |
| Kwok 2008 Hong Kong, China | HIC | Low | 78 | 45 | ≥40 | NR | Home visiting program | F-to-F | Inpatient and home-based | No | Hospital | 6 | 21 |
| Laramee 2003 USA | HIC | Some concern | 70.7 | 54 | <40 | 38 | Case management | Combined | Combination of settings | Yes | Hospital | 3 | 29 |
| Linné 2006 Sweden | HIC | Some concern | 70.5 | 70.4 | NR | NR | Primarily educational intervention | N/A | N/A (inpatient) | No | Hospital | 0.5 | 13 |
| McDonald 2002 Ireland | HIC | High | 70.8 | 66.3 | <40 | NR | Multidisciplinary care model | Combined | Combination of settings | Yes | Hospital | 3 | 27 |
| Naylor 2004 USA | HIC | Low | 76 | 42.7 | <40 | NR | Case management | Combined | Combination of settings | Yes | Hospital | 6 | 31 |
| Negarandeh 2019 Iran | LMIC | Some concern | NR | 60.3 | NR | NR | Structured telephone support | P-to-P | Telephone or internet-based | No | Hospital | 2 | 16 |
| Nucifora 2006 Italy | HIC | High | 73 | 62 | ≥40 | 64 | Case management | Combined | Combination of settings | No | Hospital | 6 | 25 |
| Ong 2016 USA | HIC | Low | 73.5 | 53.8 | ≥40 | 61.2 | Case management | Mechanized and F-to-F | Telephone or internet-based | No | Hospital | 6 | 20 |
| Pugh 2001 USA | HIC | High | 77 | NR | NR | 52.7 | Case management | Combined | Combination of settings | Yes | Hospital | 6 | 27 |
| Riegel 2002 USA | HIC | Some concern | 73.9 | 48.9 | ≥40 | 96.8 | Structured telephone support | P-to-P | Telephone or internet-based | Yes | Community | 6 | 20 |
| Riegel 2006 USA | HIC | Some concern | 72.1 | 46.3 | ≥40 | 81.4 | Structured telephone support | P-to-P | Telephone or internet-based | Yes | Community | 6 | 20 |

*(Continued)*

**Table 1.** (Continued)

| Author/Year/Country | Trial characteristics | | Characteristics of participants | | | | Characteristics of interventions | | | | | | Adapted HF-DMSI¶ |
|---|---|---|---|---|---|---|---|---|---|---|---|---|---|
| | Region* | Risk of bias | Age (years) | Male (%) | LVEF† (%) | NYHA‡ III-IV (%) | Intervention category | Communication§ | Environment | Caregiver | Place initiate | Duration of intervention (months) | |
| Ritchie 2016 USA | HIC | Low | 63.3 | 51.4 | ≥40 | NR | Case management | Combined | Inpatient and telephone-based | Yes | Hospital | 2 | 24 |
| Schwarz 2008 USA | HIC | High | 78 | 48 | NR | 79.4 | Telemonitoring | Mechanized | Telephone or internet-based | Yes | Community | 3 | 22 |
| Sethares 2004 USA | HIC | High | 76.3 | 47.1 | ≥40 | N/A | Home visiting program | Combined | Inpatient and home-based | No | Hospital | 1 | 19 |
| Stromberg 2003 Sweden | HIC | Some concern | 77.5 | 61.3 | NR | 82.1 | Clinic-based intervention | F-to-F | Clinic/outpatient setting | Yes | Community | 1 | 22 |
| Thompson 2005 UK | HIC | Some concern | 72.5 | 72.6 | <40 | 74.5 | Clinic-based intervention | Combined | Combination of settings | Yes | Hospital | 6 | 27 |
| Van Spall 2019 Canada | HIC | Some concern | 71.7 | 49.6 | NR | NR | Case management | Combined | Combination of settings | Yes | Hospital | 3 | 27 |

* Region: HIC, High-Income Country; LMIC, Low and Lower Middle-Income Country.

† LVEF, Left Ventricular Ejection Fraction.

‡ NYHA, New York Heart Association functional classification.

§ Communication: P-to-P, Person-to-Person by telephone; F-to-F, Face to Face contact individually or in a group; Mechanized, Mechanized via Internet or telephone; Combined, Combination of different communications.

¶ The adapted HF-DMSI score was used to evaluate the overall program dose (the complexity and intensity) of the interventions.

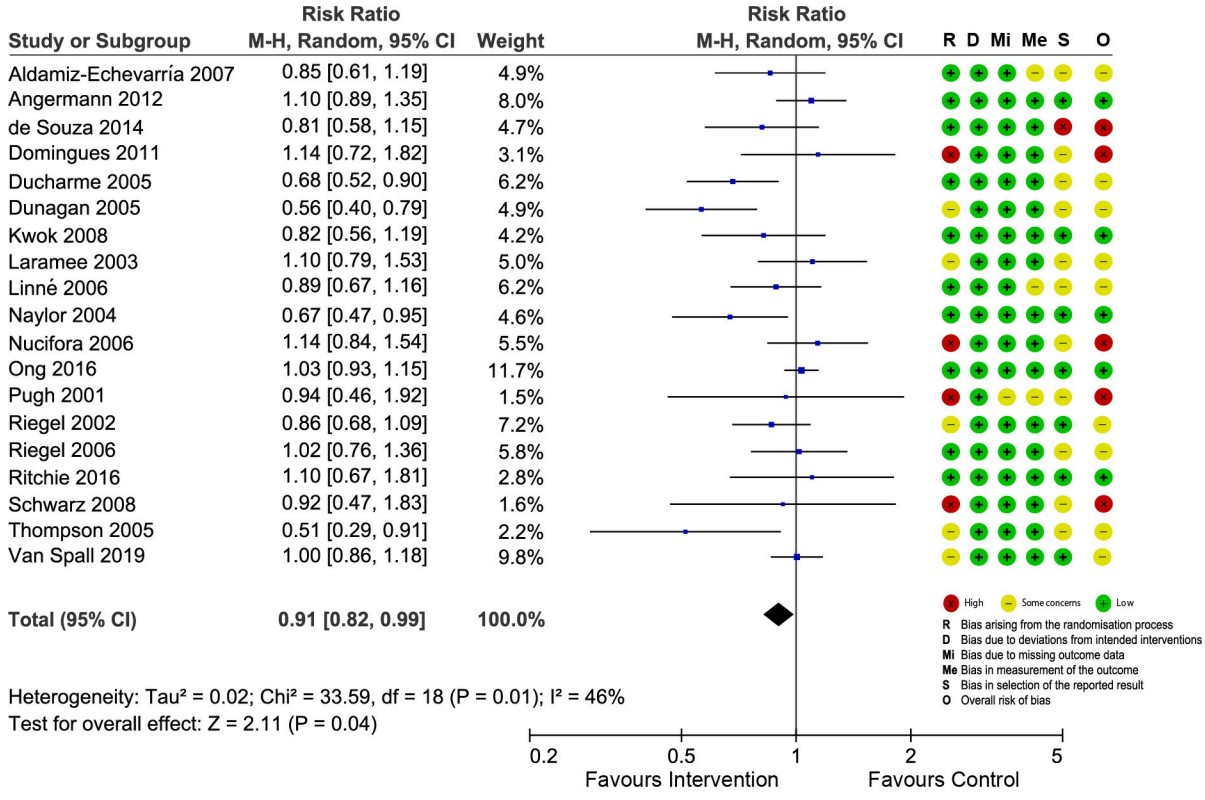

**Fig 2. Forest plot showing the effect of nurse-led TCIs on all-cause readmissions and risk of bias assessment for each study.**

Sensitivity analysis by excluding each study at a time yielded similar non-significant results for ED visits (S4 File).

**Length of hospital stay.** Five RCTs reported the effects of nurse-led TCIs on the LOS for subsequent hospitalizations (Fig 5). The combined evidence showed that nurse-led TCIs led to

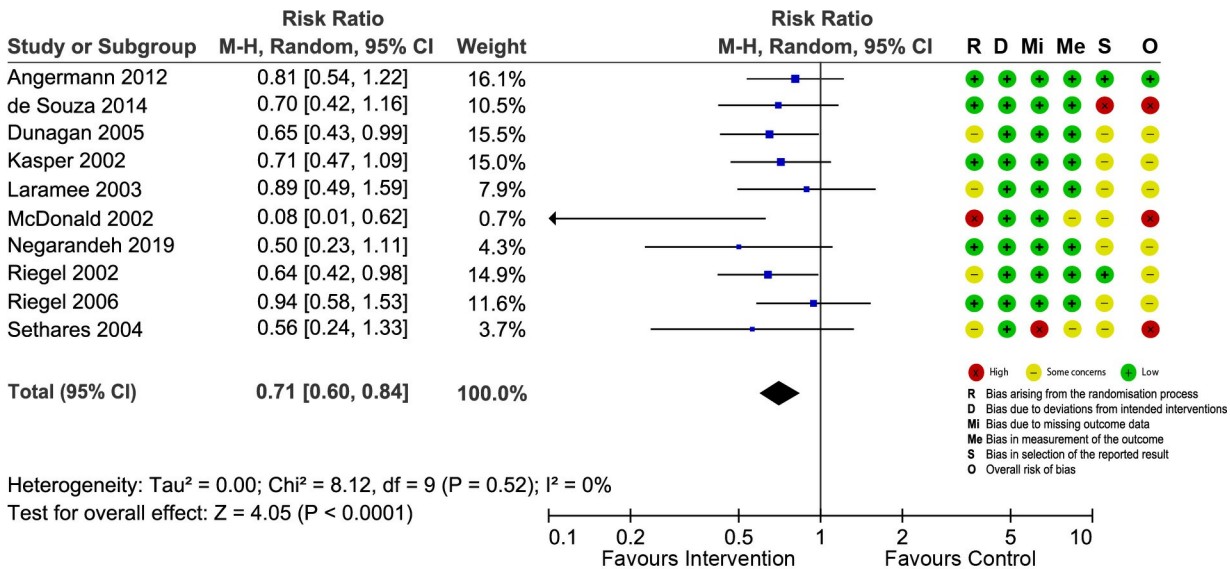

**Fig 3. Forest plot showing the effect of nurse-led TCIs on HF-specific readmissions and risk of bias assessment for each study.**

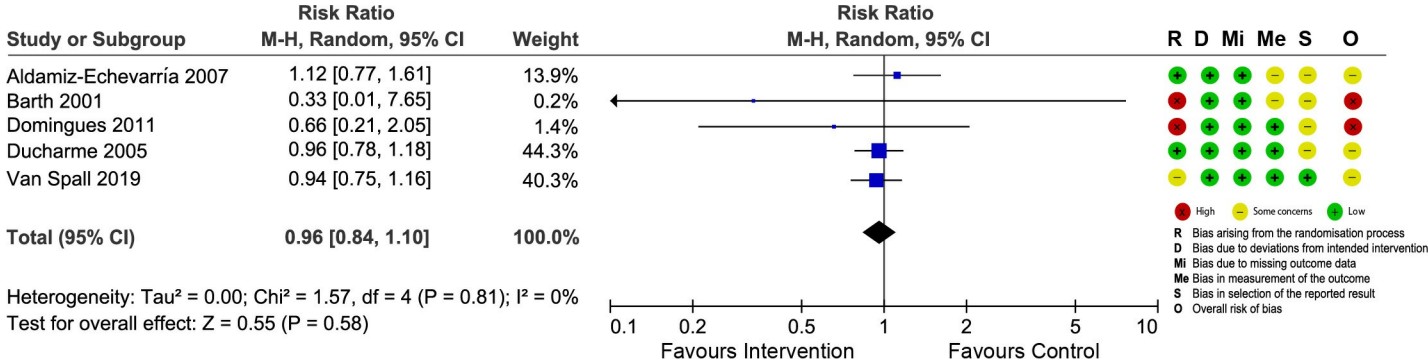

**Fig 4. Forest plot showing the effect of nurse-led TCIs on emergency department visits and risk of bias assessment for each study.**

a significant reduction in the LOS (MD = -2.37; 95% CI = -3.16 to -1.58; p < 0.0001) and the heterogeneity across studies was low ($I^2$ = 14%). Sensitivity analysis did not substantially change the summary estimates (S4 File).

**Meta-regression and dose-response meta-analysis.** Univariate meta-regression analyses assessing whether the treatment effects on all-cause and HF-specific readmissions were modified by trial characteristics, participant characteristics, as well as intervention characteristics revealed that all the prespecified parameters had little to no effect on the outcomes assessed (S5 File). Therefore, these trial-level covariates did not explain the between-study heterogeneity. The dose-response meta-analysis showed an inverse linear association between the scores of the adapted HF-DMSI and HF readmissions ($p_{linearity}$ < 0.001; Fig 6B). The overall RR trend was 0.987 (95% CI = 0.981 to 0.993). A 1.3% decrease in HF readmission risk correlated with one score increment in the "program dose" prescribed. No significant evidence of a linear dose-response relationship between "program dose" and all-cause readmissions ($p_{linearity}$ = 0.227; Fig 6A) as well as nonlinear relationships for this outcome were found ($p_{nonlinearity}$ = 0.905).

## Publication bias

The shape of the funnel plots (S6 File) did not show any evidence of asymmetry; therefore, few small-study effects across included RCTs for both all-cause and HF-specific readmissions were noted. This was further supported by non-significant p-values of Harbord's tests (all-cause readmissions: p = 0.08 and HF-specific readmissions: p = 0.06), suggesting that publication

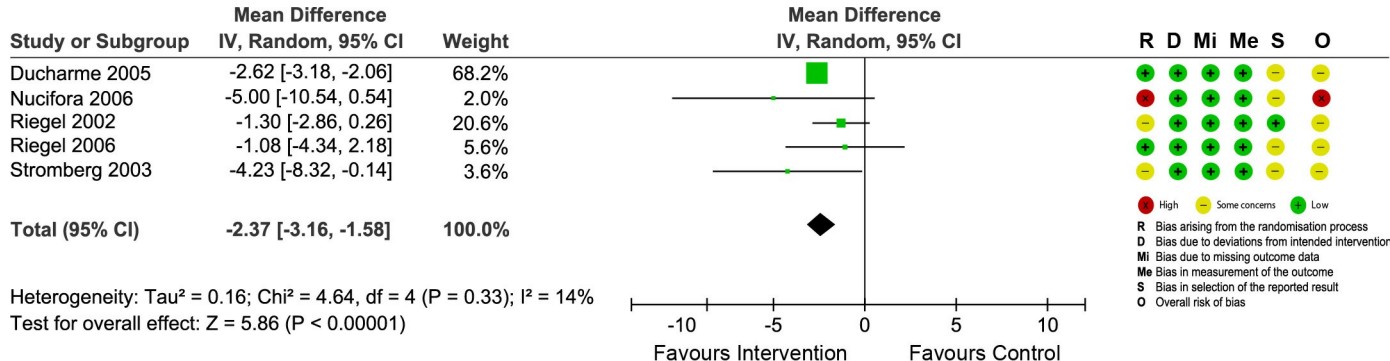

**Fig 5. Forest plot showing the effect of nurse-led TCIs on the length of hospital stay and risk of bias assessment for each study.**

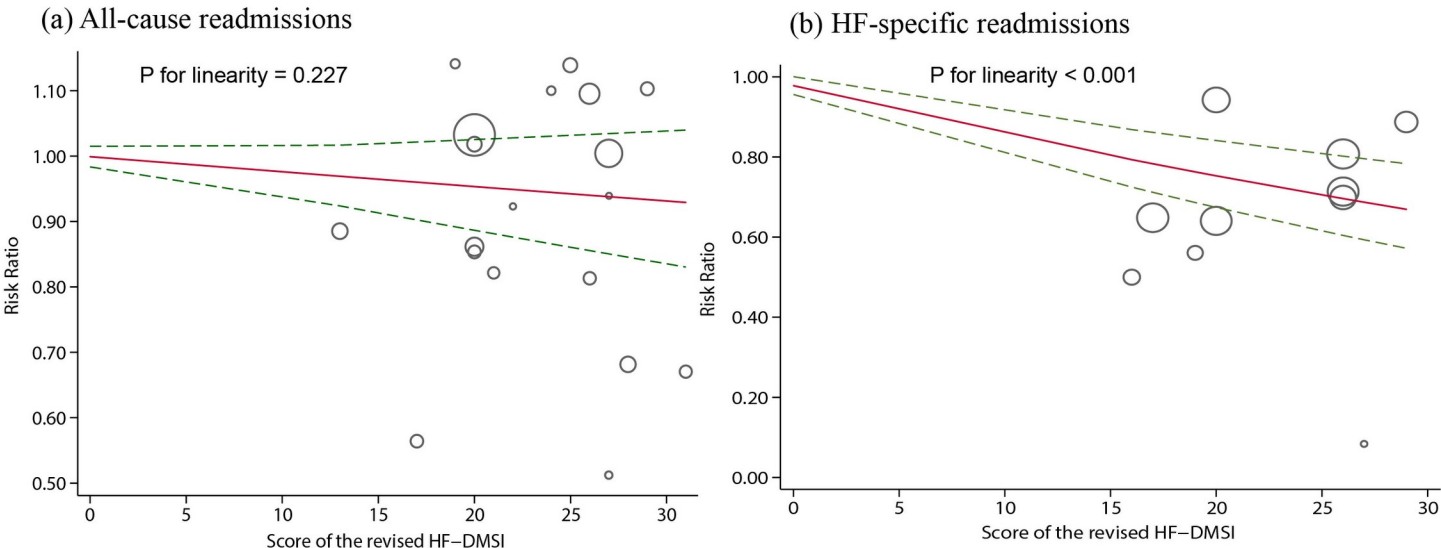

**Fig 6.** Dose-response relationship of the adapted HF-DMSI score with (a) all-cause readmissions and (b) HF-specific readmissions.

bias was unlikely to affect the results. No evidence of missing studies was found using the trim-and-fill method, which was indicated by unchanged results. Publication bias was not examined for ED visits and the LOS because limited number of studies (<10) were included in each analysis.

## Quality of evidence

The overall quality of evidence ranged from low to moderate based on the GRADE framework (S7 File). The certainty of the evidence for all-cause readmission was low because it was down-graded by two level due to the high risk of bias and inconsistency. For HF-specific readmissions, the certainty of evidence was moderate because it was downgraded by two levels due to very serious risk of bias and upgraded by one level due to the presence of a clear dose-response gradient. The certainty of evidence for both ED visits and the LOS was low because of the high risk of bias and imprecision.

## Discussion

In this review, we considered the time-limited nature of transitional care led by nurses and focused on the vulnerable period of hospital-to-home transition. As far as we know, this is the first systematic review and meta-analysis to assess the impact of nurse-led TCIs on healthcare utilization outcomes in patients with HF, using meta-regression and dose-response analyses. Pooled evidence from relevant RCTs suggested a mean 9% and 29% risk reduction for all-cause (low level of certainty) and HF-specific readmissions (moderate level of certainty), respectively, among patients with HF who received nurse-led TCIs compared with that among those who only received usual care. In addition, although nurse-led TCIs have been proven to be effective in shortening the hospital LOS (low level of certainty), they have not been found to be significantly effective in decreasing ED visits (low level of certainty).

The positive results in our study are in line with the results of a previous systematic review of studies of patients with HF discharged from hospital to home receiving transitional care services [17]. However, unlike prior review studies that generally concluded insufficient data to examine the effectiveness of TCIs on ED visits [16,17], our study provided robust and

consistent evidence that led to the conclusion that nurse-led TCIs do not contribute to the reduction of ED visits in patients with HF. One possible explanation for the difference in the findings was that some components of nurse-led TCIs assist in detecting early symptomatic deterioration and help patients who require immediate attention promptly and successfully seek emergency medical advice [50]. Moreover, concerns exist regarding financial penalties set forth by the Hospital Readmission Reduction Program for higher-than-average readmission rates that would incentivize hospitals to "game" the system by shifting inpatient-type care to EDs [51]. Nurse-led TCIs that do not improve ED visit rates can pressurize policymakers, making them cautious regarding the program funding emergency treatment or primary care [45].

We used univariate meta-regression analysis to identify effect modifiers that could potentially influence the size of the intervention effects; however, the prespecified trial characteristics, participant characteristics, as well as intervention characteristics included in our meta-regression analysis failed to modify the overall intervention effects of all-cause and HF-specific readmissions. Moreover, the association between the overall "program dose" of the nurse-led TCIs prescribed for patients with HF and risk of HF-specific readmissions appeared to follow an inverse, linear dose-response pattern, with a risk reduction of 1.3% per score increment of the intervention complexity and intensity as assessed by the adapted HF-DMSI. The result indicated that more complex and intensive nurse-led transitional care programs may lead to greater benefits in reducing HF-specific readmissions. This suggest that the complexity and intensity of intervention is likely to be an important driver of patient outcomes. This is a novel finding with important implications for improving the effectiveness of future nurse-led transitional care programs designed for patients with HF, especially when transitional care services are advocated as the standard of care for post-discharge management [52].

## Limitations and strengths

This study has a few limitations which should be mentioned. First, the bulk of the included studies were conducted in high-income countries, with almost half of them conducted in the US, and consequently, the data are insufficient to determine the extent to which these nurse-led TCIs can be applicable in low- and middle-income countries with different healthcare systems and cultures. Second, the overall risk-of-bias of the included studies was mainly judged as "high risk of bias" or "some concerns," hence, the findings should be interpreted cautiously. Third, significant amount of inconsistency in the description of usual care and interventions existed, which undoubtedly added to clinical heterogeneity and deterred from precisely capturing the overall program dose. Last, no subgroup analysis was performed on different types of nurse-led TCIs because the number of intervention types was relatively large and the number of studies for some types was too small for a reliable analysis. However, despite the aforementioned limitations, our study has provided new insights into the current state of evidence based on the most up-to-date literature without language restrictions. We tested the robustness of the synthesized results using sensitivity analysis and attempted to examine the potential sources of heterogeneity with meta-regression models considering the trial-level characteristics. In addition, the one-stage REMR method was first used in our study to model the dose-response relationship between the intensity and complexity of nurse-led TCIs and intervention effects.

## Conclusions

This systematic review focused on nurse-led TCIs that fill in the care gap from hospital to home for people with HF. Nurse-led TCIs had positive actions on all-cause readmissions, HF-

specific readmissions, and LOS. However, these interventions did not result in significant reduction in ED visits. The positive treatment effect for HF-specific readmissions was related to the intervention intensity and complexity in a dose-response pattern. We did not identify any characteristics or contexts to explain between-study heterogeneity or that favor intervention success. There is a need for future research to address the characteristics of the optimal nurse-led TCIs and the most beneficial patient population.

## Supporting information

**S1 Checklist. PRISMA 2009 checklist.**
(DOCX)

**S1 File. Search strategy.**
(PDF)

**S2 File. The adapted HF Disease Management Scoring Instrument (HF-DMSI).**
(DOCX)

**S3 File. Summary characteristics of participants and interventions in the included studies.**
(DOCX)

**S4 File. Sensitivity analysis.**
(DOCX)

**S5 File. Univariate meta-regression analysis of all-cause and HF-specific readmissions.**
(DOCX)

**S6 File.** Funnel plots for the effect of nurse-led TCIs on (a) all-cause readmissions and (b) HF-specific readmissions.
(DOCX)

**S7 File. Summary of findings.**
(DOCX)

## Acknowledgments

We would like to acknowledge Ping Xu for assistance in building search strategies and Chang Xu for providing Stata codes and statistical instructions.

## Author Contributions

**Conceptualization:** Minlu Li, Yuan Li, Yinyin Li, Jinbo Fang.

**Data curation:** Yuan Li, Qingtong Meng, Xiaomeng Tian.

**Formal analysis:** Minlu Li, Ruixia Liu.

**Investigation:** Minlu Li, Yuan Li, Yinyin Li, Xiaomeng Tian, Ruixia Liu.

**Methodology:** Minlu Li, Yuan Li, Qingtong Meng, Yinyin Li.

**Supervision:** Jinbo Fang.

**Visualization:** Yuan Li, Qingtong Meng.

**Writing – original draft:** Minlu Li, Xiaomeng Tian, Ruixia Liu.

**Writing – review & editing:** Minlu Li, Yuan Li, Qingtong Meng, Yinyin Li, Jinbo Fang.

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
