## [Decision Letter · Decision Letter 0]

25 May 2021

PONE-D-21-01523

Effects of nurse-led transitional care interventions for patients with heart failure on healthcare utilization: A meta-analysis of randomized controlled trials

PLOS ONE

Dear Dr. Fang,

Thank you for submitting your manuscript to PLOS ONE. After careful consideration, we feel that it has merit but does not fully meet PLOS ONE’s publication criteria as it currently stands. Therefore, we invite you to submit a revised version of the manuscript that addresses the points raised during the review process.

We look forward to receiving your revised manuscript.

Kind regards,

Satya Surbhi, PhD

Academic Editor

PLOS ONE

Journal Requirements:

2. Please note that PLOS ONE uses a single-blind peer review procedure. We would therefore be grateful if you could include in the information that has been redacted for peer review in the manuscript.

3. Please provide a justification for restricting your searches to studies published in the year 2000 or later.

"This work was supported by the Key Project of Science and Technology of XX Province (Province

name blinded for peer review) (No. 2020YFS0150) and the Nursing Discipline

Development Special Fund of XX (Name of university blinded for peer review) (No.

HXHL19024)."

Reviewers' comments:

Reviewer's Responses to Questions

**Comments to the Author**

1. Is the manuscript technically sound, and do the data support the conclusions?

Reviewer #1: Yes

Reviewer #2: Yes

2. Has the statistical analysis been performed appropriately and rigorously? 

Reviewer #1: Yes

Reviewer #2: Yes

3. Have the authors made all data underlying the findings in their manuscript fully available?

Reviewer #1: Yes

Reviewer #2: Yes

4. Is the manuscript presented in an intelligible fashion and written in standard English?

Reviewer #1: Yes

Reviewer #2: Yes

5. Review Comments to the Author

Reviewer #1: This manuscript is well written and comprehensive.

Only a few suggestions.

It would help to add a little more information regarding the "program dose" within the manuscript.

The studies in this review span 20 years. Any thoughts regarding change in these programs over time?

Reviewer #2: Thank you for the opportunity to review this manuscript. The topic is of continued importance, given the morbidity and cost of heart failure care, and updates prior analyses by focusing more specifically on the role of nursing in care transition interventions for hospitalized heart failure patients.

I have several recommendations that would strengthen the manuscript.

1. Please clarify the outcomes measured. I think these are all 6 month outcomes (lines 96-98) but it would be helpful to clarify this is truly the case.

2. The HF-DMSI is adapted into a COM/ty score. It is not clear what the adaptation is, including a supplemental attachment that scores the adapted HF-DMSI for the studies, presumably using some but not all of the HF-DMSI scores would be helpful for reviewers.

3. Figure 1: PRISMA: given the high number of excluded studies by title or abstract review, it might be helpful to know how many were excluded due to either category. If this was recorded, this is ok to not include.

4. Meta-regression results: the trends suggest things that I think are likely true, that these programs are better for those with HFrEF (HF with reduced ejection fraction, and that there is a publication issue that readmission reductions are likely driven by earlier findings (vs. later when EHRs and better information sharing have occurred). It also looks like there are trends related to proportion of males although not sure about the trend related to place of initiation (hospital vs. home). I would recommend reporting on these trends in the text.

5: dose response: I am not sure that I would have inferred a relationship from figure 6b, but I am glad that the figure is included and not just reporting relative risks and p values.

4. References: should be cleaned up. My spot check found that citation 26 "Inc MUaEP" on line 440 is probably truncated due to the reference management software. Citation 31 is actually published in JAMA: Internal Medicine.

6. PLOS authors have the option to publish the peer review history of their article (what does this mean?). If published, this will include your full peer review and any attached files.

Reviewer #1: No

Reviewer #2: No

---

## [Author Response · Author response to Decision Letter 0]

1 Jul 2021

Dear editor and reviewers,

Thank you for your letter dated 25 May 2021.

On behalf of my colleagues, I am herewith submitting the revised manuscript entitled “Effects of nurse-led transitional care interventions for patients with heart failure on healthcare utilization: A meta-analysis of randomized controlled trials.” (PONE-D-21-01523) for consideration of publication in PLOS ONE. We would like to thank the editor and reviewers’ work devoted to our manuscript and we are very grateful for their valuable suggestions. We have considered the comments carefully and have made revisions (highlighted in red in the revised manuscript) which we hope meet with approval.

Journal Requirements:

1. “Please ensure that your manuscript meets PLOS ONE’s style requirements, including those for file naming.”

 Response: Reviewed and revised as suggested.

2. “Please note that PLOS ONE uses a single-blind peer review procedure. We would therefore be grateful if you could include in the information that has been redacted for peer review in the manuscript.”

 Response: Thank you for your reminding. Included as suggested.

3. “Please provide a justification for restricting your searches to studies published in the year 2000 or later.”

 Response: Thank you for your rigorous comment. A nurse-led model of care called Omada program was set up around 2000 to improve the management of heart failure. Thereafter, an increasing number of nurse-led disease management interventions for patients with heart failure has been implemented. In addition, classical models of nurse-led care transitions have been established one after another since the year of 2000 with the expectation to promote healthy behavior and reduce healthcare utilization. It was the time point that people started to realize the potential role of nurses in chronic disease management for heart failure. We thus determined to restrict the searches to studies published in the year 2000 or later.

4. “Please remove any funding-related text from the manuscript and let us know how you would like to update your Funding Statement.”

 Response: Thank you for bringing this to our attention. We have removed the funding-related text from the manuscript as suggested. And we would like to update our Funding Statement in the cover letter as follows: “This work was supported by the Key Project of Science and Technology of Sichuan Province, China (Grant No. 2020YFS0150) and the West China Nursing Discipline Development Special Fund, Sichuan University (Grant No. HXHL19024). The funders had no role in study design, data collection and analysis, decision to publish, or preparation of the manuscript.”

Comments to the Author:

Reviewer #1:

1. “It would help to add a little more information regarding the ‘program dose’ within the manuscript.”

 Response: We gratefully thanks for the precious time the Reviewer #1 spent in making the positive and constructive suggestions. According to your suggestion, we have added more details about the “project dose” in lines 159-162 and 330-332 in the revised manuscript. The evaluating criterion of the “program dose” was based on the updated HF-DMSI, which was showed in S2 File.

2. “The studies in this review span 20 years. Any thoughts regarding change in these programs over time?”

 Response: We strongly agree with your valuable comment and thank you very much. We have also considered the change in these programs over time, so the year of publication across studies was included as an effect modifier in our meta-regression analysis. As a result, univariate meta-regression analyses indicated that the treatment effect sizes were not modified by our prespecified parameters, including the year of the study (S5 File).

Reviewer #2:

1. “Please clarify the outcomes measured. I think these are all 6-month outcomes (lines 96-98) but it would be helpful to clarify this is truly the case.”

 Response: We gratefully thanks for the precious time the Reviewer #2 spent in making these constructive remarks. We have revised the sentence and provided the rationale for the 6-month timeline for outcome assessment. Please see lines 97-101. 

2. “The HF-DMSI is adapted into a COM/ty score. It is not clear what the adaptation is, including a supplemental attachment that scores the adapted HF-DMSI for the studies, presumably using some but not all of the HF-DMSI scores would be helpful for reviewers.”

 Response: Thanks for your careful reading and providing us with some keen scientific insight. The supplemental attachment of the adapted HF-DMSI has been added as suggested. Please see S2 File. We have also added more descriptions about the instrument in lines 159-162 in the revised manuscript.

3. “Figure 1: PRISMA: given the high number of excluded studies by title or abstract review, it might be helpful to know how many were excluded due to either category. If this was recorded, this is ok to not include.”

Response: Thank you so much for your suggestion. We regret that we did not record the how many articles were excluded due to either titles and abstracts. But we believe that the review on the basis of title and abstract was conducted in strict accordance with the follow the inclusion criteria and exclusion criteria.

4. “Meta-regression results: the trends suggest things that I think are likely true, that these programs are better for those with HFrEF (HF with reduced ejection fraction), and that there is a publication issue that readmission reductions are likely driven by earlier findings (vs. later when EHRs and better information sharing have occurred). It also looks like there are trends related to proportion of males although not sure about the trend related to place of initiation (hospital vs. home). I would recommend reporting on these trends in the text.”

Response: Thank you for the comment and we fully understand your concerns over “trend towards statistical significance”. While we appreciate the feedback, we respectfully disagree. The paper entitled “Trap of trends to statistical significance: likelihood of near significant P value becoming more significant with extra data (BMJ 2014;348:g2215)” presents a quantitative analysis to show that describing near significant p values as “trends towards significance” (or similar) is not just inappropriate but actively misleading (undermining the principle of accurate reporting), as such p values would be quite likely to become less significant if extra data were collected. Please kindly refer to the article published in BMJ with the following link: https://www.bmj.com/content/bmj/348/bmj.g2215.full.pdf.

5. “Dose response: I am not sure that I would have inferred a relationship from figure 6b, but I am glad that the figure is included and not just reporting relative risks and p values.”

 Response: Thanks for expressing your concern regarding the figure 6b. We used the one-stage robust error meta-regression (REMR) approach within STATA to conduct the dose-response analysis and generate the corresponding figure. Following the dose-response meta-analysis, we found a statistically significant linear association between the score of the adapted HF-DMSI and HF readmissions (p linearity < 0.001). The overall trend RR was 0.987 (95% CI = 0.981 to 0.993). According to the results, we inferred an inverse, linear relationship between the scores of the adapted HF-DMSI and HF-specific readmissions. Specifically, a 1.3% decrease in HF readmission risk correlated with one score increment in the “program dose” prescribed.

6. “References: should be cleaned up. My spot check found that citation 26 "Inc MUaEP" on line 440 is probably truncated due to the reference management software. Citation 31 is actually published in JAMA: Internal Medicine.”

 Response: We are very sorry for the mistakes in the References and the inconvenience caused during your reading. We have revised all references carefully and tried to avoid citation inaccuracy. Thank you for your reminding.

The reviewers’ comments helped clarify and improve our paper. We appreciate for Editor/Reviewers' warm work earnestly.

Thank you again for your constructive comments and suggestions!

Submitted by the Authors.

---

## [Decision Letter · Decision Letter 1]

5 Oct 2021

PONE-D-21-01523R1Effects of nurse-led transitional care interventions for patients with heart failure on healthcare utilization: A meta-analysis of randomized controlled trialsPLOS ONE

Dear Dr. Fang,

Thank you for submitting your manuscript to PLOS ONE. After careful consideration, we feel that it has merit but does not fully meet PLOS ONE’s publication criteria as it currently stands. Therefore, we invite you to submit a revised version of the manuscript that addresses the points raised during the review process.

We look forward to receiving your revised manuscript.

Kind regards,

Satya Surbhi, PhD

Academic Editor

PLOS ONE

Journal Requirements:

Reviewers' comments:

Reviewer's Responses to Questions

**Comments to the Author**

1. If the authors have adequately addressed your comments raised in a previous round of review and you feel that this manuscript is now acceptable for publication, you may indicate that here to bypass the “Comments to the Author” section, enter your conflict of interest statement in the “Confidential to Editor” section, and submit your "Accept" recommendation.

Reviewer #2: (No Response)

2. Is the manuscript technically sound, and do the data support the conclusions?

Reviewer #2: Yes

3. Has the statistical analysis been performed appropriately and rigorously? 

Reviewer #2: Yes

4. Have the authors made all data underlying the findings in their manuscript fully available?

Reviewer #2: Yes

5. Is the manuscript presented in an intelligible fashion and written in standard English?

Reviewer #2: Yes

6. Review Comments to the Author

Reviewer #2: I appreciate the changes made by the authors in response to reviewer comments. With regards to the adapated HF-DMSI, Table 1 should not use in its last column header "COM/Ty" and instead use "Adapted HF-DMSI" to be consistent with the text of the manuscript. I would recommend changing the Table 1 footnote by taking out the first two words "COM/Ty: the" and capitalize "adapted" so that the footnote helps a reader understand how the HF-DMSI was adapted (and can look for more information in the text and supplement).

7. PLOS authors have the option to publish the peer review history of their article (what does this mean?). If published, this will include your full peer review and any attached files.

Reviewer #2: No

---

## [Author Response · Author response to Decision Letter 1]

15 Oct 2021

We appreciate the time the Reviewer #2 spent in making the constructive suggestions. We have revised the last column header and the footnote in the table 1 in the revised manuscript as suggested.

---

## [Decision Letter · Decision Letter 2]

1 Dec 2021

Effects of nurse-led transitional care interventions for patients with heart failure on healthcare utilization: A meta-analysis of randomized controlled trials

PONE-D-21-01523R2

Dear Dr. Fang,

We’re pleased to inform you that your manuscript has been judged scientifically suitable for publication and will be formally accepted for publication once it meets all outstanding technical requirements.

Kind regards,

Satya Surbhi, PhD

Academic Editor

PLOS ONE

Reviewers' comments:

Reviewer's Responses to Questions

**Comments to the Author**

1. If the authors have adequately addressed your comments raised in a previous round of review and you feel that this manuscript is now acceptable for publication, you may indicate that here to bypass the “Comments to the Author” section, enter your conflict of interest statement in the “Confidential to Editor” section, and submit your "Accept" recommendation.

Reviewer #2: All comments have been addressed

2. Is the manuscript technically sound, and do the data support the conclusions?

Reviewer #2: Yes

3. Has the statistical analysis been performed appropriately and rigorously? 

Reviewer #2: Yes

4. Have the authors made all data underlying the findings in their manuscript fully available?

Reviewer #2: Yes

5. Is the manuscript presented in an intelligible fashion and written in standard English?

Reviewer #2: Yes

6. Review Comments to the Author

Reviewer #2: Thank you for the changes, Table 1 now rrads appropriately with the change to the last column and the footnote.

7. PLOS authors have the option to publish the peer review history of their article (what does this mean?). If published, this will include your full peer review and any attached files.

Reviewer #2: No

---

## [Editor Report · Acceptance letter]

6 Dec 2021

PONE-D-21-01523R2 

Effects of nurse-led transitional care interventions for patients with heart failure on healthcare utilization: A meta-analysis of randomized controlled trials 

Dear Dr. Fang:

I'm pleased to inform you that your manuscript has been deemed suitable for publication in PLOS ONE. Congratulations! Your manuscript is now with our production department. 

Kind regards, 

on behalf of

Dr. Satya Surbhi 

Academic Editor

PLOS ONE